# Time-Series Chlorophyll Fluorescence Imaging Reveals Dynamic Photosynthetic Fingerprints of *sos* Mutants to Drought Stress

**DOI:** 10.3390/s19122649

**Published:** 2019-06-12

**Authors:** Dawei Sun, Yueming Zhu, Haixia Xu, Yong He, Haiyan Cen

**Affiliations:** 1College of Biosystems Engineering and Food Science, Zhejiang University, Hangzhou 310058, China; dzs0015@zju.edu.cn (D.S.); zhukvo@zju.edu.cn (Y.Z.); haixiaxu@zju.edu.cn (H.X.); yhe@zju.edu.cn (Y.H.); 2Key Laboratory of Spectroscopy Sensing, Ministry of Agriculture and Rural Affairs, Hangzhou 310058, China; 3State Key Laboratory of Modern Optical Instrumentation, Zhejiang University, Hangzhou 310058, China

**Keywords:** *Arabidopsis thaliana*, chlorophyll fluorescence imaging, drought stress, salt overly sensitive (SOS) pathway

## Abstract

Resistance to drought stress is one of the most favorable traits in breeding programs yet drought stress is one of the most poorly addressed biological processes for both phenomics and genetics. In this study, we investigated the potential of using a time-series chlorophyll fluorescence (ChlF) analysis to dissect the ChlF fingerprints of salt overly sensitive (SOS) mutants under drought stress. Principle component analysis (PCA) was used to identify a shifting pattern of different genotypes including *sos* mutants and wild type (WT) Col-0. A time-series deep-learning algorithm, sparse auto encoders (SAEs) neural network, was applied to extract time-series ChlF features which were used in four classification models including linear discriminant analysis (LDA), k-nearest neighbor classifier (KNN), Gaussian naive Bayes (NB) and support vector machine (SVM). The results showed that the discrimination accuracy of *sos* mutants SOS1-1, SOS2-3, and wild type Col-0 reached 95% with LDA classification model. Sequential forward selection (SFS) algorithm was used to obtain ChlF fingerprints of the shifting pattern, which could address the response of *sos* mutants and Col-0 to drought stress over time. Parameters including *QY*, *NPQ* and *Fm*, etc. were significantly different between *sos* mutants and WT. This research proved the potential of ChlF imaging for gene function analysis and the study of drought stress using ChlF in a time-series manner.

## 1. Introduction

World agriculture is facing the great challenge of feeding the fast-growing population, which is going to be over 9 billion by 2050, especially under the severe climate conditions, such as drought and salt stresses [1]. Drought has become a key stress factor in the production of major crops which could lead to great economical losses every year on a global scale [2]. One of the main methods to improve crop quality is through breeding programs, such as breeding elite varieties with strengthened resistance to abiotic stress, which is important for a reliable and sustainable world food supply [3]. Therefore, it is crucially important to study genes that are related to drought stress through the combination of large-scale plant phenotyping under drought stress and related gene function analysis in crop breeding programs, which could be achieved using the model plant *Arabidopsis thaliana.*

The response of plants to drought stress is one of the most poorly understood abiotic stress processes due to the complexity of drought response [4]. It could cause considerable damage to photosynthesis by introducing a perturbation in photosystem II (PSII) functional properties, which is due to the imbalance of decreased light capture ability and its utilization. During drought stress, plants usually apply various mechanisms to avoid this inhibition of photosynthesis process including preventing excessive light absorption and utilizing the reduced energy from PSII in a “power-saving” mode. These strategies are mainly achieved through regulating stomata and the biochemistry of photosynthetic metabolism. Stomatal closure is one of the early responses of plants to drought stress, which could reduce the transpiration rate, thereby improving water use efficiency. It was reported that impaired stomatal closure function could cause greater sensitivity of plants to drought stress [2]. Previous studies demonstrated one of the most important biochemical activities related to plant drought stress was oxidative stress. This, even though functions in (or catalyzing) very complex biological processes, includes superoxide radicals, hydroxyl radical, perhydroxy radical, and hydrogen peroxide, etc. Even though drought stress is not the same as salt stress, they are usually considered closely related or interdependent, because they both cause damages to plants including osmotic stress, ion toxicity, and oxidative stress [5,6]. It was reported that the salt overly sensitive (SOS) pathway is a key mechanism in preserving the homeostasis of Na^+^ and K^+^ under salt stress. SOS pathway consists of three genetic loci including SOS1, SOS2, and SOS3. SOS1 encodes a membrane protein with the size of 127 kDa that functions as a Na^+^/H^+^ antiporter which involves direct transportation of sodium ions through plasma membrane including extrude sodium ions and regulating the transportation of Na^+^ from root to shoot [7,8]. The function of the SOS2 gene is to encode a serine/threonine protein kinase in the SNF1/AMPK family, which is required for the function of maintaining intracellular Na^+^ and K^+^ homeostasis [9]. Importantly, it was demonstrated that SOS2 interacts with SOS3 by forming a SOS2/SOS3 functional subunit (also called complex) which is important in regulating the expression and/or activity of ion transporters [10]. SOS3 encodes an EF-hand type (a helix-loop-helix structural domain or motif) calcium-binding protein that contains a signature sequence for the N-Myristoylation function which is important for signaling [11]. Many genes relating to salt stress were reported to play important roles in the response of plants to drought stress, but few investigated the response of *sos* mutants against drought stress through ChlF phenotyping.

*A. thaliana* has been used widely in studies of gene function analysis and plant phenomics with biotic and abiotic stresses. Compared with using major crop species for gene function study, it is more feasible to use *A. thaliana* to investigate functional genomics and phenomics in large scale, because *A. thaliana* has the advantages of smaller size, shorter life cycle and smaller genome etc. [12,13]. Through screening large numbers of plants for genetic modified mutants, elite plant varieties with strengthened abiotic stress could be selected for agricultural production [14]. Agronomic traits such as growth rate, height, shape, color, and biomass are investigated to study the response of plants to environmental stress [15,16,17,18]. However, the traditional phenotyping methods applied by these studies usually included visual estimation and manual measurement, which were labor-intensive, time-consuming, destructive, and of inefficient particularity, especially for quantitative traits. In addition, these methods were mostly biased and unrepeatable due to human interference and inconsistent standards of phenotyping procedures applied by different research facilities. These factors have greatly limited the development of phenomics and genomic analysis [19].

Over the years, a variety of imaging technologies have been applied to successfully conduct large-scale plant phenotyping including RGB imaging (the color of each pixel is determined by the combination of the red, green, and blue intensities) [20], hyperspectral imaging [21], and infrared thermal imaging [22], which were employed to explore the impacts of biotic and abiotic stresses quantitatively and qualitatively over time. In addition to being rapid, high throughput, and simple to apply, these techniques also require minimum labor force [23], which makes imaging technologies suitable to phenotype plants in large-scale for the investigation of desirable gene functions under different environmental conditions. As one of the promising means for plant phenotyping, chlorophyll fluorescence (ChlF) imaging has been applied in a wide variety of studies to monitor physiological responses of plants to biotic and abiotic stresses [24], because not only could it measure visible changes of chlorophyll degrading caused by stress, but it also captures the subtle physiological phenomena that interfere with photosynthetic apparatus and the according biological processes [25]. Therefore, ChlF imaging could provide a comprehensive insight to understanding the fundamental mechanism of photosynthesis with both genetic and environmental changes [26]. Of the major fluorescence measurement methods, kinetic chlorophyll fluorescence imaging with quenching kinetics and light curve protocol is efficient in monitoring electron transport patterns in photosynthesis and evaluating the photosynthetic capacity [27,28]. Kinetic chlorophyll fluorescence imaging was employed in various research including the evaluation of freeze-thaw and drought stress [14], low and high light conditions [29], salt stress [30], and chronic ozone damage [31]. However, to our knowledge, there are currently no published studies that used kinetic chlorophyll fluorescence imaging to dynamically monitor various drought responses caused by certain genes (*sos* genes in this study) in a time-series manner, which could provide a novel rapid phenotyping means for analyzing gene function and possibly improve the efficiency of breeding programs.

Hence, the demand for rapid phenotyping on drought stress to benefit the progress of breeding programs and the gap between the development of phenotyping and genotyping prompted us to study the dynamic ChlF fingerprints of *sos* mutants under drought condition. Therefore, the goal of this study was to investigate the feasibility of using kinetic chlorophyll fluorescence imaging to dynamically monitor the response of *sos* mutants under drought conditions in a time-series manner and extract the ChlF fingerprints of *sos* mutants of *A. thaliana*. The findings of this research could provide an insight to understanding the functions of genes on the SOS pathway, and possibly improve the efficiency of breeding programs in the long run.

## 2. Materials and Methods

### 2.1. Plant Material and Workflow of Experiment

Accessions of *Arabidopsis* including wild type Col-0 (ecotype *Columbia*), and two *sos* mutants including SOS1-1 and SOS2-3 were provided by Plant Environmental Sensing Laboratory of Hangzhou Normal University, Hangzhou, China, which were used for drought treatment and kinetic chlorophyll fluorescence imaging protocol. Genotypes of SOS1-1 and SOS2-3 were acquired by mutagenesis procedures as described in previous research [32]. In total, 216 accessions for all three genotypes were included in this study. Each genotype had 72 replicates with three plants grown in one pot.

The experiment was carried out in the Key Laboratory of Spectroscopy Sensing (KLSS) at Zhejiang University. The flow chart of the experimental design is shown in Figure 1. The procedure of cultivating *Arabidopsis* followed a published research with minor modifications [33]. After disinfected by soaking in 70% ethanol for 1 min, 1% NaClO for 15 min and washed five times with distilled water, seeds were sown on 0.8% agar medium supplemented with 50% Murashige and Skoog salts (1/2 MS), 1.5% sucrose in Petri dishes. Plants were cultivated in an incubator (AR-4112, Percival Scientific, Perry, GA, United States) with controlled environment, in which the temperature was set at 22 °C with relative humidity at 65%. The fluency rate of cool-white fluorescent illumination was set at 100 µmol m^−2^ s^−1^ with a 16/8-h light/dark cycle. On the 15th day post sowing (four-leaf stage), plants of similar size were picked for transplanting into 210 mL plastic pots filled with mixed soil (2:1:1 for soil, perlite, and vermiculite in volume). Each pot contained three plants of each genotype. Transplanted plants were cultivated in the same environment described above with a daily routine of being watered with 6 mL 1% nutrient solution. On day 28 post transplanting, *Arabidopsis* plants reached the 10-leaf stage, when half of transplanted plants from each genotype, 108 plants in total, ceased being watered as drought treatment, while the other half were still watered daily as control. All plants were used for the following kinetic ChlF imaging process since the first day of drought treatment. The treatment lasted for 7 days. On each day during drought stress, kinetic chlorophyll fluorescence images were acquired from all samples before watering all plants of control groups. The acquired ChlF data was then used for further analysis.

### 2.2. Kinetic Chlorophyll Fluorescence Imaging

The kinetic chlorophyll fluorescence images were collected using a pulse amplitude modulated (PAM) chlorophyll fluorescence imaging system, FluoCam FC800 imaging system (Photon Systems Instruments, Brno, Czechia) in the lab of KLSS at room temperature. The system and collecting procedure were described previously [25]. The detailed setting of ChlF imaging system is depicted in Appendix A. The chlorophyll fluorescence system used a high-speed charge-coupled device (CCD) camera with a prime lens (SV-H 1.4/6, VS Technology, Tokyo, Japan) to capture the emission transients of chlorophyll fluorescence in a set of ChlF images with the spatial resolution of 696 × 520 pixels. Light source was provided by four light emitting diode (LED) panels (45° incident angle) surrounding the camera lens including two red-orange LEDs panels and two cool white LEDs panels. The red-orange LEDs panels provided flashes (<0.1 µmol m^−2^ s^−1^) and actinic light 1 (0–250 µmol m^−2^ s^−1^). An adjustable sample-loading platform was installed beneath the camera lens. The distance between samples and the lens was set at 21 cm.

The detailed quenching kinetic protocol was described as follows. Before the acquisition of ChlF images, plants were subjected to a dark adaption treatment for 15 min, which allowed the opening of PSII reaction centers. After transferring plants onto the sample-loading platform, a 4040 ms flash of light was applied to measure the minimum fluorescence (*Fo*), before the maximum fluorescence (*Fm*) was determined by imposing a saturation pulse of 2300 µmol m^−2^ s^−1^ at 5.56s for 320 ms. A constant actinic light of 100 µmol m^−2^ s^−1^ was applied for 70 s before the application of saturating flashes to acquire maximum fluorescence of light adaptation (*Fm_Ln*) at 32.24, 42.24, 52.24, and 72.24 s, corresponding to *L1, L2, L3* states, respectively. *Fm_Lss*, which was the maximum fluorescence of light adaptation at steady state, was measured at 92.24 s. *Fm_Dn* representing the instantaneous maximum fluorescence signals at dark with saturating flashes was recorded at 122.24, 152.24, and 182.24 s, corresponding to *D1, D2, D3* states, respectively. Other parameters were calculated based on the measured fluorescence signals. The ChlF parameter abbreviations, calculation formulas, and detailed quenching protocol are elucidated in Appendix A.

### 2.3. Data Analysis

The FluorCam7 (PSI version 1.1.1.5, Ltd, 1996) software was used to determine the region of interest (ROI), which was the entire leaf area of the whole plant. All the parameters of each pixel were extracted after implicating mask, subtracting background, and calculating parameters. The resulting pixels of each plant were averaged to represent the whole plant. In total, there were 98 parameters extracted from each sample which could characterize the photosynthetic condition in a comprehensive manner. The response of plant to drought stress is a complex biological process and the impact could subtly vary between plants and days. To obtain a general view of the dataset and understand the possible patterns over time under drought stress, a principle component analysis (PCA) was conducted first using all ChlF variables from the entire sample panel for 7 days. PCA could reduce variable dimensions thus providing a general interpretation in terms of possible trend pattern within the whole dataset. 

One-way analysis of variance (ANOVA) analysis was then performed to evaluate the differences of ChlF parameters between drought treatment groups. The significant differences between means were determined using Duncan test (*p* < 0.05) by SPSS (version 20, IBM Corp., Armonk, NY, USA). Sparse auto encoders (SAEs) deep learning method was then performed to dissect time-series ChlF data, which was conducted on a python program platform (Python 3.6, The Python Software Foundation) based on high-level neural networks API Keras (https://keras.io/) in PyCharm IDE (PyCharm 2018.2.2). All of the encoded and decoded activation functions were achieved with ReLU [34], which is widely used in the training of deep learning algorithms.

In order to select the most representative ChlF parameters that were related to different responses of three genotypes to drought stress, sequential forward selection (SFS) algorithm was applied to selected optimal features for each day. SFS starts with an empty set of features and selects features through a bottom-up manner which repeatedly adds the most important feature each time by the fisher criterion until the optimal number is reached with the best accuracy [25]. The selected features were then used in the Ward.D linkage clustering method to validate the performance of significant features. The ChlF data preprocessing and SFS feature selection were performed in Matlab R2014a (MathWorks, United States), while the clustering algorithm was conducted on R 3.5.2 (R Foundation for Statistical Computing, Vienna, Austria) platform. 

## 3. Results and Discussion

### 3.1. Effects of Drought Stress Based on RGB images

Figure 2a shows the RGB images of both control and treatment plants of genotype Col-0 (WT), SOS1-1, and SOS2-3 on days 1, 4, and 7 post drought treatment. It could be observed that plants of Col-0 and SOS1-1 subjected to drought stress were clearly smaller compared with those of control groups. Meanwhile, plants of Col-0 and SOS1-1 in drought treatment groups were smaller compared with those of SOS2-3, suggesting that *Arabidopsis* of Col-0 and SOS1-1 were more susceptible to drought stress than that of SOS2-3. From day 1 to day 7, plants of control groups grew bigger over time, while plants of treatment groups experienced growth retardation of different levels due to drought stress except for plants within drought groups of SOS2-3. Leaves with different degrees of wilting could be observed within treatment groups of three genotypes. No evident size difference was observed between the control group and drought treatment group of SOS2-3 plants based on RGB images, while a minor degree of wilting leaves could still be observed in drought treatment groups but not in control groups. It is reported that phenotypic changes including rolled and wilted leaves might be part of the initial mechanisms applied by plants to alleviate drought stress, since these morphological changes could decrease the reaction ratio of transpiration process therefore avoiding extra loss of water by improving water efficiency [2].

In order to gain conclusive and quantitative results of plant growth status amongst three genotypes, projected area presented by pixel are shown in Figure 2b. The plant growth of control group was not significantly different from that of treatment group on day 1 for three genotypes (*p*_Col-0_ = 0.11; *p*_SOS1-1_ = 0.20; *p*_SOS2-3_ = 0.06, respectively). On day 4, the plant growth of control group and that of treatment group for wild type Col-0 was significantly different (*p*_Col-0_ = 0.02), while no significant difference was observed for two *sos* mutants (*p*_SOS1-1_ = 0.34; *p*_SOS2-3_= 0.72, respectively). Eventually, on day 7, significant differences between the plant growth of control group and that of treatment group were observed for all genotypes (*p*_Col-0_ = 0.00032; *p*_SOS1-1_ = 0.0059; *p*_SOS2-3_= 0.03, respectively). These results indicate that *sos* mutants might be less susceptible to drought stress compared to wild type *Arabidopsis* Col-0. It is noteworthy that this result is different from a previous publication which claimed that there was no significant difference between wild type and *sos* mutants caused by drought stress [35]. In addition, the growth results suggested *sos* mutants showed arguably more robust resistance to drought stress. 

### 3.2. PCA with All ChlF Parameters and Kinetic ChlF Curves Captured Responses of sos Mutants to Drought Stress Over Time

ChlF data of three genotypes throughout the whole experimental period are recorded in Appendix A. The PCA was first conducted with ChlF data to acquire a basic overview of the dataset, and obtain a possible pattern of changes caused by genetic variations between wild type and *sos* mutants under drought stress. PCA was performed for each day throughout the experimental period using the data of all 98 ChlF parameters. During the PCA, the data dimensionality was reduced by projecting ChlF data into a principal components (PCs) space, which could decrease the problem of multicollinearity. This process was achieved by scoring each sample via PCs [36]. The 2-dimensional (2D) PC score scatter plots using the first two PCs of all the *Arabidopsis* samples from day 1 to day 6 post drought stress are shown in Figure 3. It is worth noticing that a clear clustering pattern could be observed. On day 1, three genotypes could not be clearly separated from each other, suggesting that after 24 h of drought stress there was no significant differences amongst three genotypes. On day 2, wild-type *Arabidopsis* evidently formed a distinct cluster, while two *sos* mutants could not be separated from each other, suggesting that on the second day post drought stress, the responses of *sos* mutants were different from wild type. These results indicated that *sos* genes or the SOS pathway might play a role in the response to drought stress, because the only genetic difference between wild type and *sos* mutants was the disrupted *sos* genes on the SOS pathway. From day 2 to day 4, it was most interesting to visualize an evident two clusters formed between wild type *Arabidopsis* Col-0 and *sos* mutants (day 3) generally shifting into three distinct clusters amongst three genotypes (day 4). This indicated that even SOS1-1 and SOS2-3 both had a malfunctioned SOS pathway, genes *sos*1 and *sos*2 might have different roles to play in the process of responding to drought stress which could include regulating other biological processes or signaling other pathways [37,38,39] involved in drought responses. From day 5 to day 6, no clear clusters could be observed, indicating that, to this point, drought might have caused enough stress on all genotypes that no difference could be observed based on the ChlF parameters. 

In order to gain a better understanding of photosynthetic performance of three genotypes over time based on kinetic ChlF imaging, ChlF curves of three genotypes from day 1 to day 7 were obtained. Based on the results of PCA analysis, ChlF curves of three genotypes on days 2, 3, and 4 were compared in Figure 4. It could be clearly observed that most difference occurred within *Fm* and *Fm_Ln* parameters for all three genotypes on days 2, 3, and 4. The ChlF curve of SOS2-3 did not change as much as those of the wild type Col-0 and SOS1-1 mutant. It is worth noticing the up shifting of the *Fm* and *Fm_Ln* region on the ChlF curve of SOS1-1 from day 2 to 4. On day 2, the values of *Fm* and *Fm_Ln* region of SOS1-1 were higher compared with those of wild type and SOS2-3 (Figure 4a). On day 3 the values of this region decreased, and a cross occurred for SOS1-1 and WT between the *Fm_L1* and *Fm_L2*. For SOS1-1, the values of *Fm* and *Fm_Ln* regions before this cross were the lowest amongst three genotypes, while the values of *Fm* and *Fm_Ln* regions after this cross were in the middle of those of wildtype and SOS2-3 (Figure 4b). In addition, eventually on day 4, the cross went down to *Fm_L3* and *Fm_L4*, and more parameters in *Fm* and *Fm_Ln* regions were lowest of the three genotypes (Figure 4c), indicating the ChlF parameters’ down-shifting process of the SOS1-1 mutant over time caused by drought stress.

### 3.3. Time-Series Deep-Learning Algorithm Classification Based on ChlF Imaging

Time-series data, which is repeated measurements over time, could provide a comprehensive and systematic understanding of biological processes. It especially facilitates studies which investigate the relationship between genotype and phenotype to shed light on gene function [40]. To comprehensively understand and differentiate three genotypes using the ChlF data from day 1 to day 7, SAEs was applied to dissect the ChlF data by evaluating the classification results. The average of training epochs for each spectral was set to 200. The training loss stopped decreasing dramatically after 50 epochs, so 200 epochs was enough for our study (Appendix A). In the SAEs neural network, ChlF data matrix was first input and transformed by the auto-encoders as vectors in the shape of (1 × 98) (98 indicates all 98 variables acquired by ChlF imaging process). After the down sampling process of the hidden layers, the original vector was decreased to (1 × 60), (1 × 40), (1 × 20), (1 × 10), and (1 × 5), respectively to further extracting the optimal features. The decoding layers reconstructed the final vector (1,5) to (1,98) in the reverse way as the input vectors under 200 epochs training with feedback propagation. After SAEs, the encoded ChlF features were used for machine learning classification, and we applied four classic classification models including linear discriminant analysis (LDA), k-nearest neighbor classifier (KNN), Gaussian naive Bayes (NB), and support vector machine (SVM). The result is shown in Table 1. It could be observed that after analyzing ChlF data in the deep-learning algorithm in a time-series manner the classification results of all four commonly used classification models reach over 90% accuracy. Of LDA, KNN, NB, and SVM algorithms, even though NB and SVM have higher training accuracy scores of 97.8% and 98.8%, respectively, their according validation accuracy was lower compared to other algorithms (90.0%, 93.3% of NB and SVM compared with 95.0%, 91.7% for LDA and KNN respectively). Overall, LDA provided a better and more robust result for the accuracy of both training dataset and validation dataset with an accuracy of 96.7% and 95.0%, respectively. This result is comparable to some other research that using ChlF imaging as a tool for abiotic and biotic stresses [2,25]. Furthermore, by combining data across several time points (7 days) post treatment, the deep-learning might be able to capture all possible subtle phenotypic variations and changes that could otherwise not be detected by classification using data from a single time point, especially for complex variation-over-time datasets [41]. Even though SAEs with time-series data combined with machine learning classification algorithms could generate accurate classification algorithms, the biological meanings of extracted features (coded features by the SAEs algorithm) could not be addressed nor understood by human beings. Therefore, a feature selection method should be employed to extract ChlF features that could be used to address and dissect the biological processes involved in response to drought stress.

### 3.4. SFS Feature Selection as Fingerprints for Responses of sos Mutants to Drought Stress

The response of plants to drought stress is complex and it is therefore one of the most poorly addressed abiotic stress processes, especially when it occurs over a time period [4]. Many biological processes including morphological and physiological changes might get involved to facilitate plants’ defense against drought stress [42], which might occur on different days post drought stress. Therefore, it is more appropriate to analyze drought stress in a time-series manner instead of just using the data from one particular day or the earliest day that a significant difference was first observed. The results of PCA analysis and kinetic chlorophyll fluorescence curves clearly indicated that there was a pattern amongst three genotypes in response to drought stress over time. However, the results are not conclusive. In addition, key variables of all 98 parameters contributing to the shifting patterns of three genotypes in response to drought stress over time needed to be addressed.

In order to selected optimal ChlF features for each day throughout the experimental period, the SFS feature selection algorithm was employed to select features that could represent the responses of three genotypes caused by drought stress for each day post drought stress. Table 2 lists the results of the SFS feature selection method. From day 0 to day 7, different features were selected for each day indicating the complexity of response to drought stress for three genotypes and that many different biological processes which could affect the photosynthesis process might be involved over time.

Selected parameters were then used in the Ward.D linkage clustering algorithm to validate the optimal parameters that contributed most to the drought response of three genotypes. Since the most noteworthy pattern shifting occurred from day 2 to day 5, the clustering results of days 2–5 are shown in Figure 5 for better visualization. Comparable results could be observed using selected parameters as that of PCA using all ChlF variables, because the same pattern was observed in the result of the Ward.D linkage clustering algorithm. On day 2, only wild type Col-0 was completely separated from *sos* mutants, yet SOS1-1 could not be isolated from SOS2-3, which agreed with the results of PCA (Figure 5a). On day 3, Col-0 plants were all clustered into cluster 1, and SOS2-3 were all within clusters 2 and 3, indicating that wild type and SOS2-3 were successfully classified into different groups, but SOS1-1 plants exist within all three clusters (Figure 5b). On day 4, three genotypes were distinctly categorized into three clusters indicating selected features could identify plants of each genotype on day 4 (Figure 5c). On day 5, three genotypes could not be separated from each other (Figure 5d). This pattern agreed with the results of PCA discussed above, suggesting the selected features contributed significantly to the identified pattern of photosynthesis performance post drought stress represented by ChlF parameters. This shifting pattern over time by selected features could be applied as fingerprints to dissect phenotypic changes of *sos* mutants to drought response in a time-series manner, and be used for the future analysis.

### 3.5. Impacts of Drought Stress on ChlF Parameters Over Time

ChlF parameters including commonly used parameters and SFS selected features are shown in Figure 6 to investigate the ChlF related plant response to drought stress over time. Commonly used parameters included *Fo, Fm, Fp, Ft_Lss, Rfd_Lss* and *Fv/Fm* (Figure 6a), while 12 of representative SFS selected parameters are shown in Figure 6b and c. For a better visualization, true values of all variables were transferred by dividing the average of all replicates from one genotype by overall average value of each feature. It could be observed in Figure 6 that for many traits of both commonly used and SFS selected features, evident differences could be identified mainly through days 3 and 4, which agrees with the PCA and Ward.D clustering results. 

Drought stress greatly affects the capacity of plant photosynthesis by causing changes in transpiration and metabolism [2]. Therefore, ChlF imaging could be applied to monitor the performance of photosynthesis by evaluating photochemical and non-photochemical changes. No significant differences over time were observed for commonly used parameters *Fm* and *Fp*, while *Rfd_Lss* of three genotypes were significant for the whole experiment. *Fo* and *Ft_Lss* showed significant difference on day 4 which agrees with previous discussion. The values of *Fv/Fm* were significantly different amongst three genotypes indicating that it might be a good indicator for plant drought stress. However, this result was opposite to the previous report that *Rfd* was more sensitive to drought stress compared with *Fv/Fm* [43]. Significant differences are shown in Appendix A. For SFS selected features, quenching coefficients of the dark-adapted state including *NPQ* and *ΦNPQ* were mostly affected except for *ΦNPQ_Lss*, while quenching parameters of light adapted state including *qP*, *QY, Fm* were evidently affected, which might be part of the reason why these traits were selected by SFS. *Fm* indicates the maximum fluorescence in light, which could reflect the electron transporting in PSII [2]. *Fm (Fm_L4* and *Fm_Lss)* of SOS2-3 increased dramatically compared to Col-0 and SOS1-1 which only increased from day 1 to day 2 before it went down again (Figure 6b). This suggests that SOS2-3 might have more maximum fluorescence capacity which could be activated to increase photosynthesis efficiency, therefore increasing the resistance to drought stress. Instantaneous PSII quantum yield QY could indicate the effects of induced damage on the photosynthetic machinery [44]. *NPQ_Lss* could be applied to evaluate stimulated electron flow and dissipated excess excitation energy as heat in the process of PSII protection effect, which indicates the adaption of plant to drought stress [45]. Photochemical quenching coefficient *qP* represents the reaction of open PSII reaction centers [23]. The *qP_L1* from day 2 to day 4 increased dramatically coincided with the decrease of *Φ**NPQ_Lss* during this time for *sos* mutants, indicating more open PSII reaction centers and less heat dissipation (Figure 6c). This might suggest that the *sos* mutants are responding to drought stress by decreasing the waste of energy thus increasing the efficiency of photosynthesis. The mechanisms of *NPQ* represent the change of H^+^, which could not only indicate the subtle fluctuation of pH gradient during PSII, but also links to the energy consumption of plants in response to drought stress by the dark reactions and energy dissipation in the light-harvesting PSII. Since the *sos* genes are responsible for the Na^+^/H^+^ exchange for the homeostasis, it agrees with the fact that most of the *NPQ* parameters *NPQ* and *ΦNPQ* were selected as ChlF fingerprints. In addition, *QY* (also known as *ΦPSII*) parameters are mainly used to evaluate the photochemistry efficiency of PSII, which is representing the proportion of light harvested by chlorophyll of PSII for photochemistry. The significant decrease of selected *QY* parameters including *QY_L1, QY_L3, QY_Lss* and *QY_L4* (Figure 6b,c) of Col-0 indicated that more sensitivity to drought stress caused severe stomatal closure which reduced the CO_2_ supply to chloroplasts [2]. The detailed meaning of selected parameters and the summary of statistics of all parameters are listed in Appendix A, respectively.

However, many traits did not show significant differences, including *ΦNPQ_D2, ΦNPQ_D3* and *qP_D2* and traits that were not selected by SFS, indicating the importance of feature selection in analyzing ChlF data and it might be more suitable for a time-series multivariate analysis. Shifting patterns varied for different commonly used and SFS selected traits, which indicated the complexity of plant responses to drought stress over time. Therefore, it might be challenging to dissect subtle changes of ChlF phenotypic traits over time (such as *qP_D2, Fv/Fm* and *Φ**NPQ_D3*), especially for different genotypes that are generated by mutation of single genes. As a consequence, using selected features for a deep learning method could be used in a time-series manner to decrease the bias and difficulty in analyzing ChlF data over time. Figure 7 shows the representative chlorophyll fluorescence images based on the ChlF variable *Fm_Lss.* The heterogeneities and temporal variations of whole plants of three genotypes could be observed. As the drought stress increases slowly over time, the values of *Fm_Lss* decreased for Col-0. *Fm_Lss* of SOS1-1 increased from day 2 to day 3 before it went down. No significant decrease could be identified from day 2 to day 4 for SOS2-3, indicating a stronger resistance to drought stress compared with the other two genotypes. 

Although analysis of ChlF parameters could provide a better understanding of drought response on each day and a pattern could be observed based on the selected variables, it is still difficult to apply classification algorithms to differentiate plants of different genotypes and link genotype to phenotype to understand a certain gene function [46,47]. This is because the response of plants to drought stress is complex, and many different biological processes or genetic regulations caused by abiotic stress were involved which might last for days, especially for chronic drought stress. In addition, plant response to drought stress could also be affected by the experimental environment, genotypes and setting of ChlF systems etc. These factors would increase the difficulties in addressing phenotypic variations based on ChlF parameters of a single day, which therefore, could not provide a comprehensive and systematic way of scoping the biological process and phenotypic variations that were usually subtle for analysis of on one time point. 

## 4. Conclusions

In this study, we investigated the potential of using a time-series ChlF analysis in dissecting ChlF fingerprints of *sos* mutants under drought stress. A clear shifting pattern was observed from the results of PCA using all acquired ChlF variables over time. The time-series deep-learning algorithm, SAEs neural network, was applied to extract time-series ChlF features which were then used in four classification models including LDA, KNN, NB, and SVM. The results showed that ChlF fingerprints could differentiate SOS1-1, SOS2-3, and Col-0 with the accuracy of 95% using the LDA classification model. After SFS selection, the same pattern was observed with the Ward.D linkage clustering algorithm. The selected parameters including *NPQ, ΦNPQ, qP, QY,* and *Fm* at different states could be applied to address the subtle yet complex changes of three genotypes over time. It was observed that in this study, *sos* mutants did not show compromised drought resistance compared to wild type Col-0, which was opposite to the reported research that suggested no significant differences in drought resistance were caused by *sos* genes. SOS2-3 mutant showed most resistance to drought stress of three genotypes, which could be observed by selected ChlF features. This research was the first attempt to dissect ChlF related phenotypic traits of mutants generated by a single gene mutation in a timer-series manner, which proved the potential of ChlF imaging for gene function analysis and the study of drought stress using ChlF in a time-series manner.

## Figures and Tables

**Figure 1 sensors-19-02649-f001:**
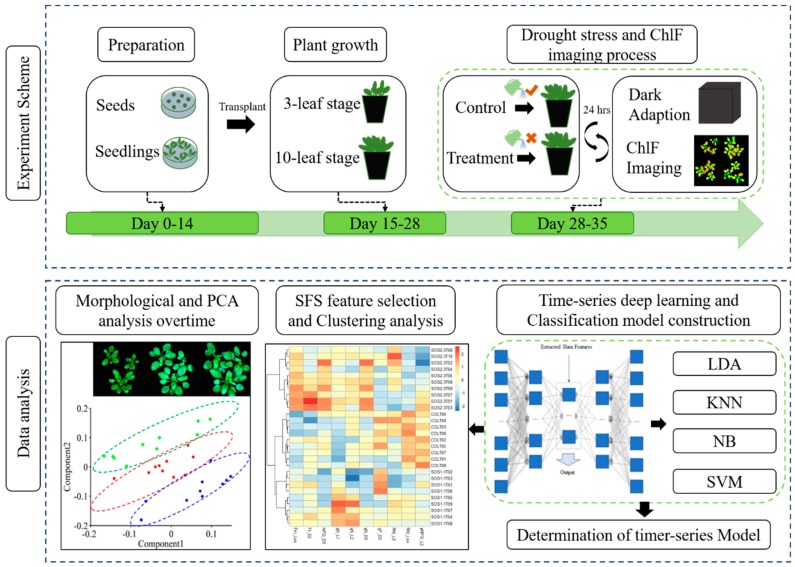
Workflow of using chlorophyll fluorescence imaging to dynamically monitor photosynthetic fingerprints caused by *sos* genes under drought condition. LDA: linear discriminant analysis; KNN: k-nearest neighbor classifier; NB: Gaussian naive Bayes; SVM: support vector machine.

**Figure 2 sensors-19-02649-f002:**
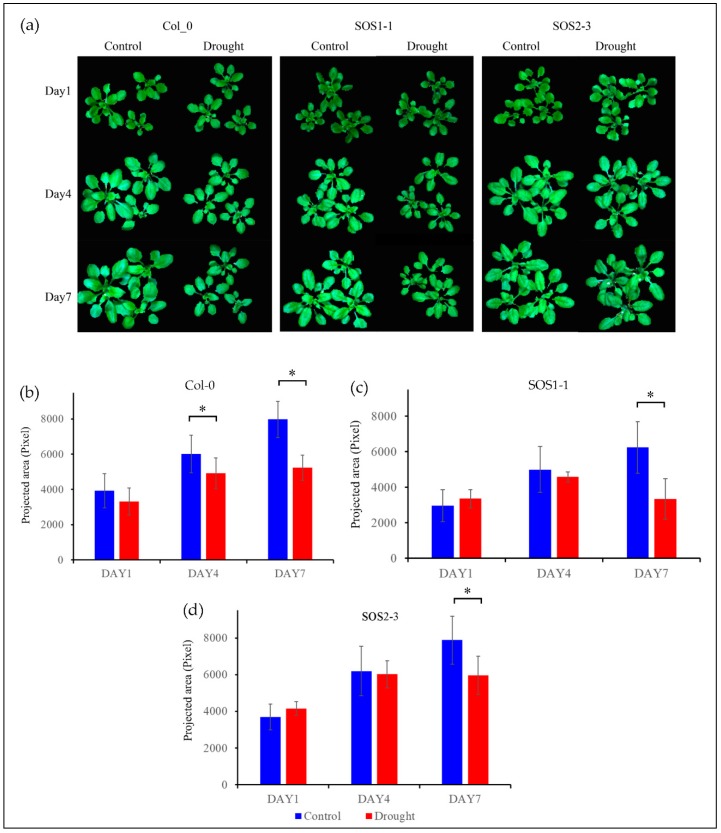
RGB images (The color of each pixel is determined by the combination of the red, green, and blue intensities) and growth status of three genotypes based on projected area presented by pixel. (**a**) RGB images of control groups and drought treatment groups of *Arabidopsis thaliana* of three genotypes including wild type Col-0, SOS1-1, and SOS2-3 on days 1, 4, and 7 respectively. (**b**) Wild type Col-0 plant growth of both control and drought treatment groups on days 1, 4, and 7 respectively. (**c**) Mutant SOS1-1 plant growth of both control and drought treatment groups on days 1, 4, and 7 respectively. (**d**) Mutant SOS2-3 plant growth of both control and drought treatment groups on days 1, 4, and 7 respectively. * represents the significant differences between indicated groups as tested with a one-way analysis of variance (ANOVA, *p* < 0.05).

**Figure 3 sensors-19-02649-f003:**
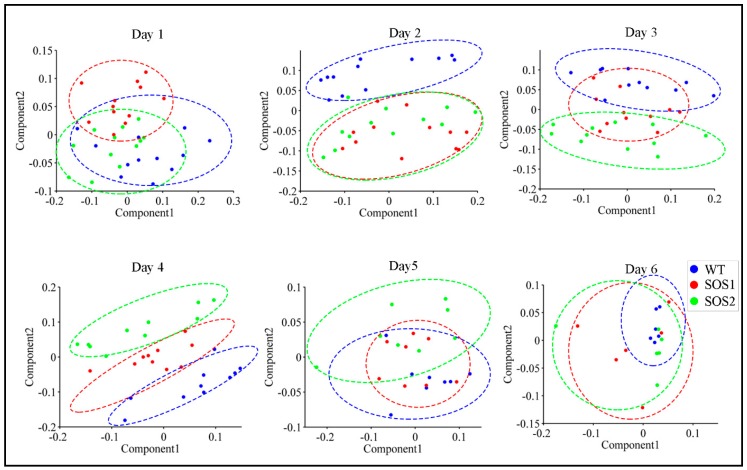
Principal component analysis (PCA) of all chlorophyll fluorescence features to visualize the trend from day 1 to day 6. The scatter plots were based on PCA scores of the first two principle components (PCs). Blue dots indicate samples of wild type Col-o; red dots indicate samples of *sos* mutant SOS1-1; green dots indicate samples of *sos* mutant SOS2-3; the dashed colored clustering circles are the same colors as those of sample dots.

**Figure 4 sensors-19-02649-f004:**
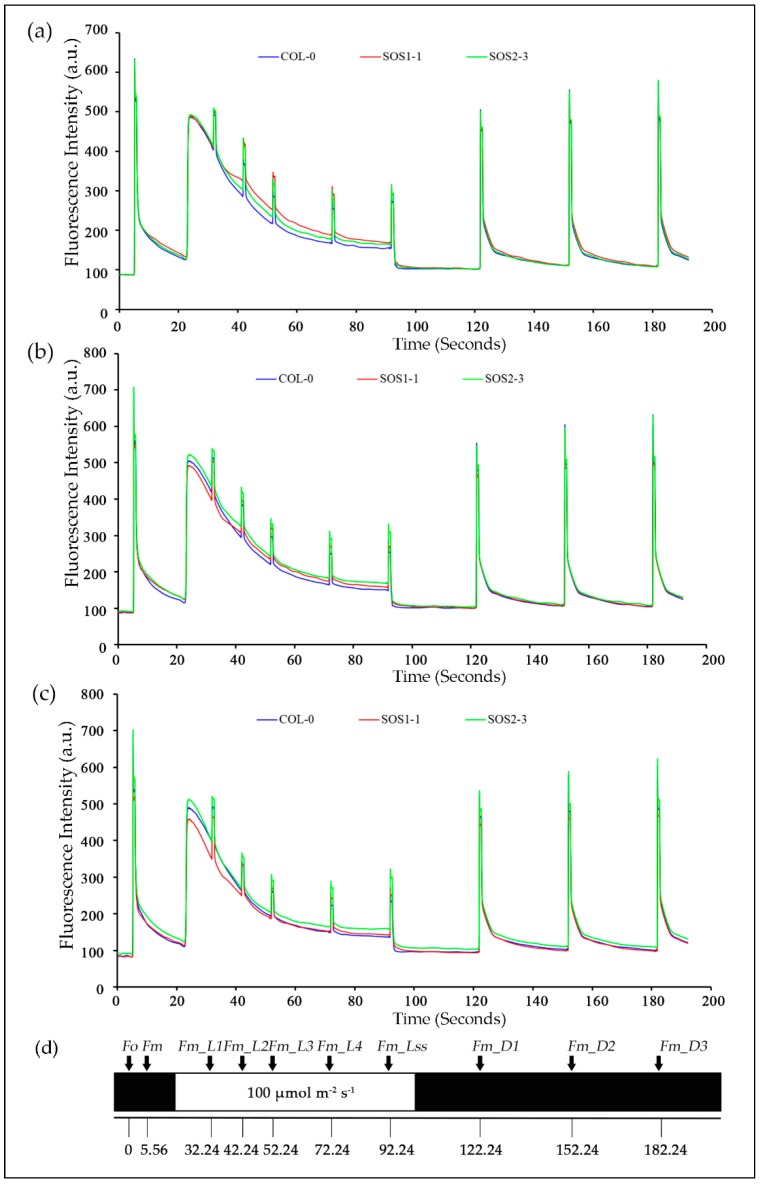
Mean kinetic chlorophyll fluorescence curves of three genotypes on days 2, 3, and 4, respectively. (**a**) Mean kinetic chlorophyll fluorescence curves of three genotypes on day 2. (**b**) Mean kinetic chlorophyll fluorescence curves of three genotypes on day 3. (**c**) Mean kinetic chlorophyll fluorescence curves of three genotypes on day 4. Blue lines indicate average kinetic chlorophyll fluorescence curves of wild type Col-o; red lines indicate average kinetic chlorophyll fluorescence curves of *sos* mutant SOS1-1; green lines indicate average kinetic chlorophyll fluorescence curves of *sos* mutant SOS2-3. (**d**) The schematic quenching protocols.

**Figure 5 sensors-19-02649-f005:**
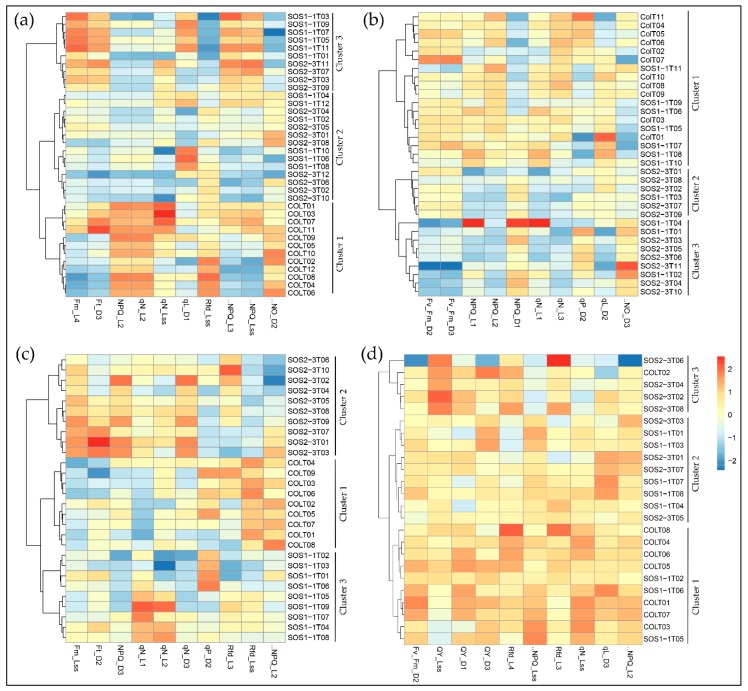
Clustering of ChlF parameters using the Ward.D linkage algorithm with selected features on (**a**) day 2, (**b**) day 3, (**c**) day 4, and (**d**) day 5.

**Figure 6 sensors-19-02649-f006:**
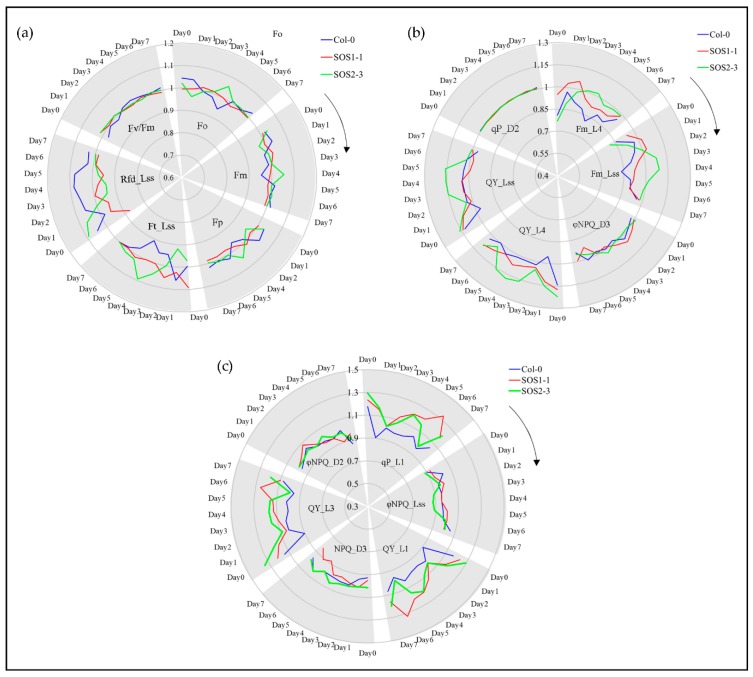
ChlF-related responses of wild type (WT) and *sos* mutants to drought stress based on selected representative features in a time-series manner. (**a**) Changes of commonly used ChlF parameters including *Fo, Fm, Fp, Ft_Lss, Rfd_Lss* and *Fv/Fm* for WT and *sos* mutants to drought stress over time. (**b**) Changes of representative selected ChlF parameters including *Fm_L4, Fm_Lss, ΦNPQ_D3, QY_L4, QY_Lss* and *qP_D2* for WT and *sos* mutants to drought stress over time. (**c**) Changes of representative selected ChlF parameters including *qP_L1, ΦNPQ_Lss, QY_L1, NPQ_D3, QY_L3* and *ΦNPQ _D2* for WT and *sos* mutants to drought stress over time. Blue lines indicate average ChlF values of wild type Col-o; red lines indicate average ChlF values of *sos* mutant SOS1-1; green lines indicate average ChlF values of *sos* mutant SOS2-3.

**Figure 7 sensors-19-02649-f007:**
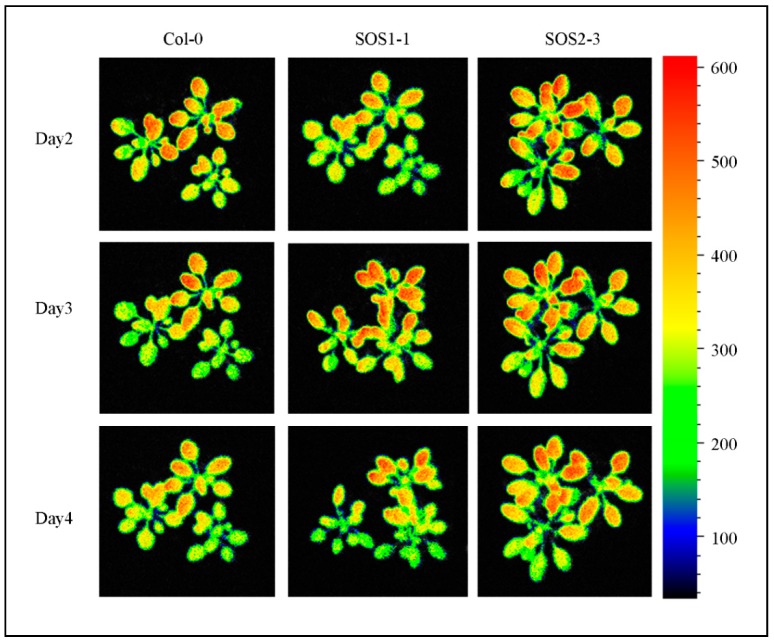
Images of representative chlorophyll fluorescence feature *Fm_Lss* of wild type Col-0, SOS1-1 and SOS2-3 *Arabidopsis* plants on days 1, 4 and 7 post drought stress.

**Table 1 sensors-19-02649-t001:** The classification results after deep-learning in a time-series manner.

Method	Training Accuracy (%)	Validation Accuracy (%)
LDA	96.7	95.0
KNN	96.7	91.7
NB	97.8	90.0
SVM	98.8	93.3

**Table 2 sensors-19-02649-t002:** Optimal features selected for each day throughout the experimental period by sequential forward selection (SFS).

Days	Selected ChlF Features									
DAY0	*Fv_Fm_L3*	*Fv_Fm_Lss*	*Rfd_L1*	*Fv_Fm_L4*	*qN_D2*	*NPQ_D3*	*ΦNO_L4*	*qL_L4*	*NPQ_D2*	*qP_L2*
DAY1	*qN_L3*	*qN_Lss*	*Rfd_L2*	*QY_Lss*	*qN_L1*	*ΦNPQ_D3*	*QY_D1*	*NPQ_L2*	*NPQ_L1*	*ΦNO_L1*
DAY2	*NPQ_L2*	*qN_L2*	*Fv_Fm_L1*	*qL_Lss*	*qL_D1*	*ΦNPQ_D2*	*Rfd_L1*	*ΦNPQ_L3*	*qP_L1*	*ΦNPQ_D3*
DAY3	*NPQ_L2*	*qN_L2*	*Fv_Fm_D2*	*Fv_Fm_D1*	*qP_D1*	*ΦNPQ_Lss*	*NPQ_D3*	*QY_D3*	*qL_D1*	*ΦNO_D2*
DAY4	*Rfd_L4*	*qP_D1*	*ΦNPQ_D2*	*qN_L1*	*Fm_L4*	*Rfd_L2*	*Ft_D1*	*NPQ_L1*	*NPQ_D3*	*qN_D2*
DAY5	*Rfd_L4*	*NPQ_Lss*	*QY_D1*	*QY_D3*	*Fv_Fm_D2*	*QY_Lss*	*Rfd_L3*	*qN_Lss*	*qL_D3*	*ΦNPQ_L2*
DAY6	*NPQ_L1*	*QY_D2*	*QY_D1*	*NPQ_D2*	*qL_L1*	*Fm_L3*	*NPQ_D3*	*qL_D3*	*Rfd_Lss*	*Fv_Fm_Lss*
DAY7	*NPQ_L4*	*qL_Lss*	*NPQ_D2*	*Rfd_L1*	*Rfd_L2*	*Ft_D2*	*qP_D3*	*QY_L2*	*QY_L4*	*-*

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
