# Peer review of "Time-Series Chlorophyll Fluorescence Imaging Reveals Dynamic Photosynthetic Fingerprints of sos Mutants to Drought Stress"

_sensors, 2019, doi:10.3390/s19122649_

Round 1
Reviewer 1 Report
The manuscript submitted by Sun et al.: "Time-series chlorophyll fluorescence imaging reveals dynamic photosynthetic fingerprints of sos mutants to drought stress" deals with interesting idea of using dynamic measurements of chlorophyll fluorescence kinetics (imaging) to discriminate between WT and sos mutants which differ in drought tolerance. The authors compared several classification methods which mostly showed high potential for detecting differences in drought tolerance, at least the one connected with sos gene. The manuscript is well organized and the results are presented clearly. The major concern I have is, that the paper is mostly about the use of different classification algorithms, without trying to understand the changes in chlorophyll fluorescence parameters during drought stress and their relation to drought tolerance. Particularly by using only three genotypes/mutants it is crucial to try more understand how the PSII photochemistry is affected by drought stress and how the dynamic differs between drought sensitive and tolerant genotype.
I would recommend to try extract the relationships between drought response of genotype (relative change of leaf area) and changes in individual chlorophyll fluorescence parameters by PCA or better RDA analysis where the direction and strength of associations between tha major drought tolerance trait (leaf area) will be compared with individual chlorophyll fluorescence parameters. I dont understand why PCA is used to demonstrate clustering of genotypes when more proper discrimination methods exist (e.g. PLS-DA) and later the authors use different classification algorithms. On the other hand PCA or RDA are very usefull tools to evaluate the relationships between individual morphological and chlorophyll fluorescence traits, the effect of drought stress and genotypes. This would help understand which chlorophyll fluorescence parameters are closely associated with drought response/tolerance.
The introduction is well writenand organized, but I am missing the paragraph related to effect of drought stress on PSII photochemistry and also brief introduction to stomatal and biochemical limitations of photosynthesis under drought stress.
In the Materials and Methods please consider merging of 2.1 and 2.2 sections, because in the first you are introducing some words about the pots, soil type (mixture) and climate conditions, but the details are in the secon section and this is not easy to follow. There are also some unnecessary data on lines 157-158 which indicate the range of intensities of actinic and saturating light of the instrument but finaly only one intensity was used.
For readers not familiar with SAE, the numbers shown in lines 309-311 in brackets need some explanation.
The Table 1 is redundant as the same data are presented in the text.
The english is relatively good but there are numerous typos e.g. l. 247 genetypes, l. 269 differnce, l. 276 muant, l. 454 patter. Therefore I recommend to read the manuscript carefully to avoid these errors.
In generall, I recommend tto accept he manuscript after minor revisions, which include particularly the attepmt to clarify more the importance of individual chlorophyll fluorescence parameters in drought tolerance detection by chaging the PCA or using RDA for individual fluorescence parameters a the main indicator of drought tolerance - leaf area.
Author Response
Please refer to the uploaded PDF file for the point-by-point response to the reviewer 1’s comments

Reviewer 2 Report
The scheme of the experiment and the model plant is described in detail in the abstract. I miss a more detailed description of the results. I believe that it would be appropriate to add the measured values to the abstract. The introduction includes and describes the issue well. The methods are sufficient. It describes the methods. I recommend adding more characteristics of the substrate. It is not entirely clear whether the duration of stress was sufficient. It measured soil water potential and RWC? I recommend adding results to the results. The results are just a description of the data obtained. I also personally miss the discussion. Charts 2-3 are difficult to read.
Author Response
Please refer to the uploaded PDF file for the point-by-point response to the reviewer 2’s comments.
